# *Know Me, Respond to Me:* Benchmarking LLMs for Dynamic User Profiling and Personalized Responses at Scale

**Bowen Jiang**[1]*, **Zhuoqun Hao**[1]*, **Young-Min Cho**[1], **Bryan Li**[1], **Yuan Yuan**[1],
**Sihao Chen**[2], **Lyle Ungar**[1], **Camillo J. Taylor**[1†], **Dan Roth**[1†]
University of Pennsylvania, Philadelphia, PA[1]
Microsoft, Redmond, WA[2]
{bwjiang, zhuoqunh, jch0, bryanli, yyuan86}@upenn.edu
sihaochen@microsoft.com, {ungar, cjtaylor, danroth}@upenn.edu

## Abstract

Large Language Models (LLMs) have emerged as *personalized* assistants for users across a wide range of tasks – from offering writing support to delivering tailored recommendations or consultations. Over time, the interaction history between a user and an LLM can provide extensive information about an individual's traits and preferences. However, open questions remain on how well LLMs today can effectively leverage such history to (1) internalize the user's inherent traits and preferences, (2) track how the user profiling and preferences evolve over time, and (3) generate personalized responses accordingly in new scenarios.

In this work, we introduce the 😊 PERSONAMEM benchmark. PERSONAMEM features curated user profiles with over 180 simulated user-LLM interaction histories, each containing up to 60 sessions of multi-turn conversations across 15 real-world tasks that require personalization. Given an *in-situ* user query, i.e. query issued by the user from the first-person perspective, we evaluate LLM chatbots' ability to identify the most suitable response according to the current state of the user's profile. We observe that current LLMs still struggle to recognize the dynamic evolution in users' profiles over time through direct prompting approaches. As a consequence, LLMs often fail to deliver responses that align with users' current situations and preferences, with frontier models such as GPT-4.1, o4-mini, GPT-4.5, o1, or Gemini-2.0 achieving only around 50% overall accuracy, suggesting room for improvement. We hope that PERSONAMEM, along with the user profile and conversation simulation pipeline, can facilitate future research in the development of truly user-aware chatbots. Code and data are available at github.com/bowen-upenn/PersonaMem.

## 1 Introduction

In recent years, Large Language Models (LLMs) have rapidly evolved as general task solvers, demonstrating remarkable performance (Srivastava et al., 2023; Zhou et al., 2023; Yue et al., 2024; Rein et al., 2024). Today, many users rely on LLMs as their *personalized* chatbots or assistants in a wide range of daily tasks – from offering writing support (Mysore et al., 2024; Tian et al., 2024) to delivering recommendations (Hua et al., 2023) or consultations (Xie et al., 2024a; Zheng et al., 2024), etc. Personalization in LLMs involves adapting model responses to specific traits, preferences, and historical interactions of each user, moving beyond generic responses to more relevant and tailored ones. Since different users have different personas, it becomes an emergent need for LLMs to be *pluralistic*—capable of adapting to different user characteristics across different scenarios (Sorensen et al., 2024; Jiang et al., 2024; Xie et al., 2024b; Kirk et al., 2024), thereby enhancing user experience and engagement.

---

*Equal contribution
†Equal advising

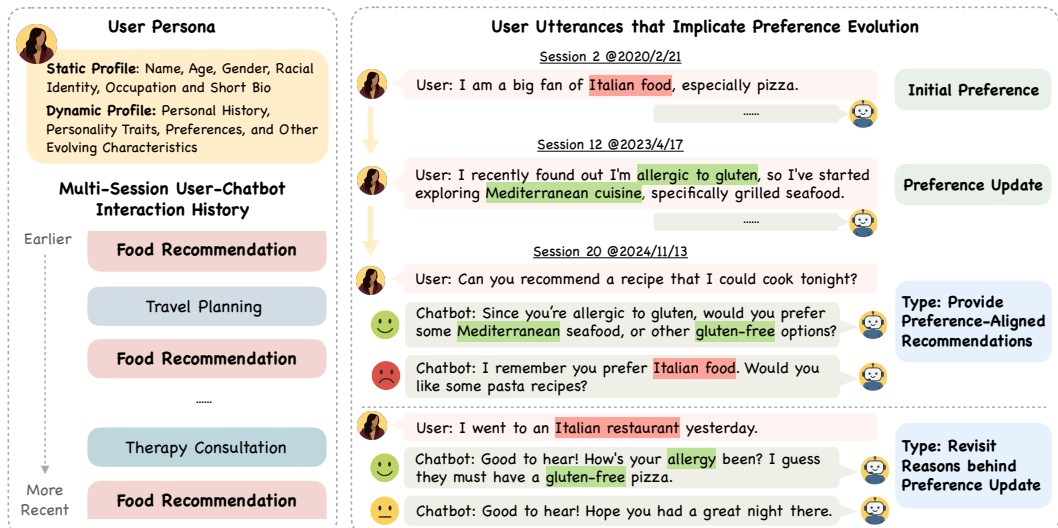

Figure 1: Overview of PERSONAMEM benchmark. Each benchmark sample is a user persona with static (e.g., demographic info.) and dynamic attributes (e.g., evolving preferences). Users engage with a chatbot in multi-session interactions across a variety of topics such as food recommendation, travel planning, and therapy consultation. As the user's preferences evolve over time, the benchmark offers annotated questions assessing whether models can track and incorporate the changes into their responses.

For LLMs to deliver personalized responses, a practical challenge lies in the fact that LLMs cannot easily access all the information about a user. This challenge is further amplified by the *ever-changing* nature of user preferences over time (Radlinski & Craswell, 2017; Dean & Morgenstern, 2022). For example, as illustrated in Figure 1, a user initially said, *"I like pizza"*, but mentioned in a later session, *"I've started exploring gluten-free options,"* upon discovering a gluten allergy. When the user again asks for food recommendations, a personalized LLM chatbot should be able to track the change, and provide recommendations according to the user's current situation. Current LLM chatbots often fail to recognize and adapt to evolving user personas. This may lead users to perceive these chatbots as less helpful and empathetic, ultimately diminishing satisfaction (Aggarwal et al., 2023; Ait Baha et al., 2023).

In this work, we evaluate LLMs' ability to leverage the *past interaction history* with a user in order to deliver a personalized response in real time. Recent studies (Lin et al., 2024; Shi et al., 2024; Zhao et al., 2025) have found that user-LLM interactions can be a rich (but often implicit) information source on the user's characteristics and preferences. However, it remains an open question whether LLMs can effectively use the interaction histories to (1) internalize the user's inherent traits and preferences, (2) track how the user's characteristics evolve over time, and (3) generate personalized responses accordingly in new scenarios.

To study these questions, we propose the 🧑 PERSONAMEM benchmark, comprising over 180 simulated user-LLM interaction histories with up to 60 multi-turn sessions across 15 personalized task scenarios. Each history is built from a detailed user persona whose characteristics evolve over time. Based on the user's profile at different points, we simulate task-specific conversations (e.g., travel, therapy, food) and concatenate them in temporal order to capture the user's profile evolution throughout the entire interaction history.

With 🧑 PERSONAMEM, we evaluate whether state-of-the-art LLMs can infer evolving user profiles and generate personalized responses across task scenarios. To emulate the realistic settings in user-LLM interactions, we design 7 types of *in-situ* user queries (Table 1), where users issue queries to LLMs from first-person perspectives. We evaluate whether LLMs can select the correct response that best aligns with the current state of the user. We find that frontier models such as GPT-4.1, o4-mini, GPT-4.5, o1, or Gemini-2.0-Flash score only around 50% overall accuracy and Llama-4-Maverick slightly lower at 43% using direct prompt approaches. While models perform reasonably well on recalling facts and tracking

preference changes (60–70% accuracy), they struggle to incorporate users' latest situations into responses (30–50% accuracy). We provide detailed analysis on how factors such as history length, preference positioning, and memory components may impact performance.

To summarize our key contributions and findings:

- We propose the 🧑 PERSONAMEM benchmark and its synthetic dialog generation pipeline for persona-oriented, multi-session, and timelined user-chatbot interaction history.

- We assess 15 LLMs on 7 types of *in-situ* user queries and evaluate their ability to provide responses aligned with user's dynamically changing profile across 15 task scenarios.

- With PERSONAMEM, we observe that frontier models such as GPT4.1, o4-mini, GPT-4.5, o1, DeepSeek-R1, Gemini-2.0, Llama-4, and Claude-3.7 still struggle to be user-aware and deliver personalized responses, especially when the knowledge of the user needs to be applied across new scenarios.

## 2  🧑 PERSONAMEM Benchmark: Overview

We present an overview of the PERSONAMEM benchmark in Figure 1. Each instance in the benchmark dataset features a *user profile or persona*, which includes basic demographic information (such as name, age, gender, and occupation), as well as *dynamic* user characteristics such as user traits, preferences, and events happening in the user's life. The dynamic user characteristics change over time as different events happen to the user that will lead to changes in users' traits and preferences specific to each task scenario.

At different points in time of a user's profile evolution, the user engages in multi-turn conversations with LLM and seeks help or suggestions from LLM on one of the task scenarios. In each task scenario, the user would ask for the LLM's suggestions given the user's need and current situation. The conversation sessions across different tasks are interleaved by the temporal order in which the sessions happen.

To understand how well LLM chatbots can track the evolution in a user's profile from the conversation histories, we evaluate LLMs by whether they can provide the most suitable response to *in-situ* user queries, where the user issues the query to LLM in a new conversation session from the first-person perspective. Depending on the time of the *in-situ* query, the expected response from the model will differ. We cast the problem as a multiple-choice selection, where LLM needs to identify the correct response out of four choices, where the incorrect choices are based on either outdated or irrelevant information with respect to the current state of the user's profile.

**Types of skills evaluated.**  To evaluate LLMs' ability to (1) memorize the user profile, (2) track how the user profile evolve over time, and (3) generate personalized responses accordingly in new scenarios, we design the following 7 types of *in-situ* user queries in the PERSONAMEM benchmark. We include examples for each type of user queries in Table 1.

1. **Recall user-shared facts.** We evaluate whether a personalized chatbot can recall static events, activities, or interests the user has shared in previous interactions, and incorporate the information in its responses.

2. **Suggest new ideas.** We evaluate whether a chatbot can suggest new items or activities that have not been mentioned in the interaction history, when users explicitly request so, e.g. "*suggest new restaurants I haven't ordered from before*".

3. **Acknowledge latest user preferences.** We evaluate whether a chatbot can recognize the latest preference expressed by the user in the interaction history.

4. **Track full preference evolution.** We evaluate whether a chatbot can keep track of how users' preferences shift by time.

5. **Revisit reasons behind preference updates.** We evaluate whether a chatbot can recall the reason(s) or event(s) leading to the preference change from a user.

6. **Provide preference-aligned recommendations.** We test whether a chatbot can proactively offer new recommendations that aligns with the user's current preferences.

7. **Generalize to new scenarios.** We evaluate whether a chatbot can transfer what it learns about the user from other task scenarios to a new task.

**Benchmark data statistics.** 🧑 PERSONAMEM features 20 personas, with over 180 interaction histories. Each interaction history contains 10, 20, or 60 sessions, where we dynamically adjust the total length of the history to approximately 32$k$, 128$k$, and 1$M$ tokens, respectively. Each session consists of 15–30 conversation turns between a user and an LLM chatbot. The user-LLM conversations span across 15 diverse topics, ranging from therapy and legal advice to recommendations on books, music, movies, and food; personal matters such as family, dating, health, and finance; and practical tasks like travel planning, online shopping, studying tips, and home decoration. In total, the benchmark features around 6$k$ *in-situ* user query and LLM response pairs across the 7 query types. Detailed dataset breakdown is discussed in Appendix D. The size of our benchmark is not limited by the scalability of the synthetic data pipeline but to make the evaluation cost reasonable.

| Query type | Examples of *in-situ* user queries and chatbot responses |
|---|---|
| **[1] Recall user-shared facts** | *"User: I shared my playlist with my friends and they loved it. ...... (later) User: What are some creative ways to share music? — Chatbot: Curating personalized playlists can be fun."* |
| **[2] Suggest new ideas** | *"User: ...... (later) User: Suggest new restaurant I haven't ordered before."* |
| **[3] Acknowledge latest user preferences** | *"User: I am a big fan of Italian food. ...... (later) User: I prefer Mediterranean cuisine now. ... (later) User: I went to an Italian restaurant yesterday — Chatbot: Nice to hear! Though I thought you prefer Mediterranean food now."* |
| **[4] Track full preference evolution** | *"User: ...... (later) User: I am thinking about attending a cooking class. — Chatbot: I understand your perspective on cooking classes has evolved. Initially, you expressed a dislike, but later you mentioned your interest in them."* |
| **[5] Revisit reasons behind preference updates** | *"User: I am a big fan of Italian food. ...... (later) User: I recently found I'm allergic to gluten, so I've shifted to other dishes. ... (later) User: I went to an Italian restaurant yesterday — Chatbot: How's your allergy been? I guess they must have a gluten-free pizza."* |
| **[6] Provide preference aligned recommendations** | *"User: I prefer Mediterranean seafood now. ...... (later) User: How can I make impactful lifestyle changes? — Chatbot: How about diving into the Mediterranean diet plan?"* |
| **[7] Generalize to new scenarios** | *"User: I prefer my yoga class because of its fixed weekly schedule. ...... (later) User: I recently haven't had time to cook. — Chatbot: Since you seem to love routine would you be interested in weekly scheduled bento-box deliveries?"* |

Table 1: Examples of the 7 types of *in-situ* user queries and expected chatbot response in the 🧑 PERSONAMEM benchmark. We highlight words that signal the user's facts or preferences.

# 3 Constructing Examples in 🧑 PERSONAMEM At Scale

We develop a modular data curation pipeline—powered by GPT-4o (Hurst et al., 2024)—to synthesize persona-oriented, multi-session user–model conversations with long context. The pipeline minimizes irrelevant or randomly injected content to better evaluate how effectively LLM chatbots address the challenges outlined in Section 2, while ensuring cost-effectiveness and scalability: generating data for each persona on each conversation topic costs approximately $2, independent of the context window length up to 1$M$ tokens.

**Construct user profile and persona.** We sample a set of random personas from PersonaHub (Ge et al., 2024), each comprising about one to three sentences, and augment them with additional demographic information and extended personal details. We also construct a timeline and populate it with events that align with the persona. These events serve as

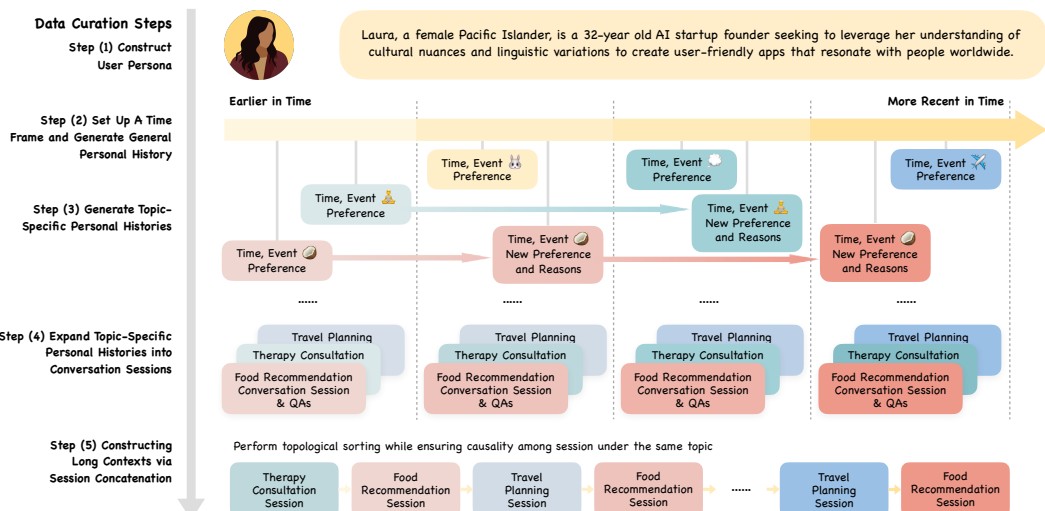

Figure 2: **An overview of the persona-oriented multi-session data curation process**. We construct user personas, build time-stamped general and topic-specific personal histories, expand them into conversation sessions, and topologically concatenate sessions to create long conversation contexts—resulting in a scalable generation framework.

the *general personal history*, such as education, career development, and life experiences, to provide a richer context. The prompts used in the process can be found in Appendix G.

Building on the persona and general personal history, we generate one additional *topic-specific personal history* for each conversation topic. Under each topic, we define a set of initial preferences, ensuring no overlap across different topics. Each topic-specific history includes events, timestamps, associated preferences, potential updates to those preferences, and the underlying reasons for those changes. This approach ensures a coherent progression of user experiences while maintaining a strong connection to their personas.

The structured personal histories also facilitate the curation of question–answer pairs. We leverage short-form information within these histories to extract ground-truth user profiles and preferences at any specific time, ensuring that the correct answers are both event- and persona-grounded. In contrast, distractor options, while generally reasonable, either overlook the user's persona or contradict it. Additionally, we exclude all questions that the model can answer correctly without seeing any contextual information from the benchmark.

**Simulate conversation sessions from user profile.** We divide the timeline into multiple segments, resulting in segments of personal histories that follow a causal, chronological order. Each segment is then expanded into a full user–model conversation session, designed to cover all details of the corresponding topic-specific personal history segment, together with additional storytelling context as if the user is talking with a chatbot naturally. For example, under the therapy consultation topic, we frame the interaction as a user seeking guidance from an AI therapist.

To enhance the quality of the conversations, we incorporate several tricks: (1) Before generating each user–model interaction turn, we prompt GPT-4o to first identify and cite the relevant event from the personal history. These citations serve as internal guidance and are not included in the final evaluation data. (2) Since GPT-4o may miss some events, leading to incomplete preference update sequences, we employ a self-reflection mechanism. We ask GPT-4o to review the generated conversation and identify any missing events from the personal history, ensuring better coverage and coherence across the interaction.

**Assemble interaction history via session concatenation.** Generating large-scale, persona-oriented long-context conversations can be both *cost-efficient* and *scalable*. For each persona, we topologically sort conversation sessions based on their ending timestamps, and we only

need to make sure sessions within the same topic maintain causality. *Different numbers of sessions can be concatenated in multiple valid orders.* This flexible design allows for multiple valid interleavings of sessions across different topics, meaning *we only need to generate sessions themselves—not every entire long-context conversation from scratch.* To further extend context length and simulate more natural interactions, we insert a limited number of short interactions between sessions where the user asks random knowledge questions or programming helps without indicating any user preferences.

**Human validation on dataset quality.**   To evaluate the quality of our generated data, we conduct a human study on 90 random query–response pairs from PERSONAMEM, each grounded in user persona, personal histories, and associated utterances in conversation. Three annotators assess each Q&A pair across four dimensions: appropriateness, relevance, correctness, and best response. Judgments were very high for all dimensions – 97.8%, 95.6%, 97.8%, and 90.0% respectively. Further details are provided in Appendix B.

## 4   Experiment

### 4.1   Evaluation Settings

Given an *in-situ* user query and the user's interaction history up to a point in time, we evaluate models' ability to select the most appropriate response according to the current state of the user amonst four different choices. Only one of the choices fits the user's current status, and the other choices contain either irrelevant or outdated facts or preferences from the user. During evaluation, apart from the conversation history, the models have access to the basic demographic information of the user, including name, age, gender identity, racial identity, and occupation. The models do not have direct access to the user's other dynamic characteristics and personal history otherwise.

For selecting the most appropriate response, we evaluate models under both discriminative and generative settings. In the *discriminative setting*, the models are presented with all four response choices denoted with (a), (b), (c) and (d) with random ordering among the choices. The model is asked to output the correct choice along with a brief explanation. In the *generative setting*, the models still see one question at a time. We compute the log-sum of token probability of generating each option individually with length normalization, and select the option with the highest probability as the model response. We use the *discriminative setting* for main evaluation (§ 4.2,§ 4.3, § 4.4) and adopt the *generative* setting in § 4.5, as it requires access to logits over entire vocabulary during decoding, which is not available from most proprietary models. No LLM judges are involved in the evaluation process.

### 4.2   Evaluating Language Models in Long-Context Settings

We first evaluate language models in the long-context setting, where the full user-LLM interaction history is provided as input to the models. Due to the length of the history, all models here were evaluated zero-shot, without demonstration examples of other histories and user queries. Our evaluation covers GPT-4.1, o4-mini, o3-mini, GPT-4.5, o1, GPT-4o, GPT-4o-mini, Gemini-2.0-Flash, Gemini-2.0-Flash-Lite, Gemini-1.5-Flash, DeepSeek-R1-671B, Llama-4-Maverick, Llama-3.1-405B, Claude-3.7-Sonnet, and Claude-3.5-Haiku (OpenAI, 2025a; 2024b; 2025b; 2024a; Hurst et al., 2024; Team et al., 2024; Guo et al., 2025; Grattafiori et al., 2024; Anthropic, 2024) on 128$k$-token context windows. We also evaluate models that support longer contexts—Llama-4-Maverick, Gemini-2.0-Flash, Gemini-2.0-Flash-Lite, and Gemini-1.5-Flash—on 1$M$-token context windows. We report the following findings:

**GPT-4.5, GPT-4.1, and Gemini-1.5 achieve the highest overall performance.**   Among leading foundation models, GPT-4.5 and Gemini-1.5 outperform others in overall accuracy. However, their performance still hovers around 52% in a multiple-choice setting, highlighting substantial room for improvement. **Notably, reasoning models such as o1, o3-mini, o4-mini, and DeepSeek-R1-607B do not demonstrate competitive advantages over non-reasoning models in the personalization tasks we evaluate.**

| Query Type \ Model | Gemini 1.5-Flash | GPT-4.5 | GPT-4.1 | o1 | Gemini 2.0-Flash | o4-mini | Gemini 2.0-Flash-Lite | GPT-4o | DeepSeek R1-671B | Llama 4-Maverick | o3-mini | GPT 4o-mini | Llama 3.1-405B | Claude 3.5-Haiku | Claude 3.7-Sonnet | Average | Random Guess |
|---|---|---|---|---|---|---|---|---|---|---|---|---|---|---|---|---|---|
| Revisit Reasons Behind Preference Updates | 0.77 | 0.76 | 0.84 | 0.75 | 0.79 | 0.75 | 0.77 | 0.77 | 0.83 | 0.76 | 0.72 | 0.70 | 0.41 | 0.64 | 0.57 | 0.72 | 0.25 |
| Tracking Full Preference Evolution | 0.65 | 0.68 | 0.67 | 0.67 | 0.70 | 0.73 | 0.68 | 0.68 | 0.68 | 0.66 | 0.54 | 0.60 | 0.38 | 0.55 | 0.45 | 0.62 | 0.25 |
| Recall User Shared Facts | 0.54 | 0.61 | 0.65 | 0.50 | 0.50 | 0.42 | 0.49 | 0.41 | 0.43 | 0.37 | 0.47 | 0.55 | 0.38 | 0.29 | 0.25 | 0.46 | 0.25 |
| Acknowledge Latest User Preference | 0.59 | 0.55 | 0.50 | 0.54 | 0.52 | 0.55 | 0.51 | 0.46 | 0.42 | 0.43 | 0.39 | 0.34 | 0.31 | 0.27 | 0.09 | 0.43 | 0.25 |
| Provide Preference Aligned Recommendations | 0.55 | 0.44 | 0.57 | 0.42 | 0.51 | 0.41 | 0.52 | 0.37 | 0.49 | 0.42 | 0.41 | 0.41 | 0.37 | 0.32 | 0.20 | 0.43 | 0.25 |
| Generalize Reasons to New Scenarios | 0.54 | 0.46 | 0.53 | 0.39 | 0.46 | 0.38 | 0.33 | 0.32 | 0.38 | 0.32 | 0.30 | 0.33 | 0.21 | 0.20 | 0.29 | 0.36 | 0.25 |
| Suggest New Ideas | 0.15 | 0.27 | 0.19 | 0.25 | 0.15 | 0.17 | 0.16 | 0.24 | 0.16 | 0.20 | 0.11 | 0.10 | 0.20 | 0.06 | 0.28 | 0.18 | 0.25 |
| Overall Accuracy | 0.52 | 0.52 | 0.52 | 0.50 | 0.49 | 0.48 | 0.48 | 0.45 | 0.45 | 0.43 | 0.39 | 0.39 | 0.31 | 0.30 | 0.26 | 0.43 | 0.25 |

Figure 3: Evaluation results across different models on 7 *in-situ* query types. We observe models perform reasonably well at recalling user facts and preferences. However, models struggle at providing novel suggestions, or applying users' preferences in new scenarios.

| Model \ Num of Sessions | Overall | 1 | 2 | 3 | 4 | 5 | 6 | 7 | 8 | 9 | 10 | 11 | 12 | 13 | 14 | 15 | 16 | 17 | 18 | 19 (128k tokens) | 20 |
|---|---|---|---|---|---|---|---|---|---|---|---|---|---|---|---|---|---|---|---|---|---|
| Gemini-1.5-Flash | 0.52 | 0.74 | 0.56 | 0.53 | 0.54 | 0.50 | 0.47 | 0.49 | 0.51 | 0.53 | 0.50 | 0.49 | 0.42 | 0.54 | 0.48 | 0.44 | 0.53 | 0.65 | 0.57 | 0.48 | 0.48 |
| GPT-4.5 | 0.52 | 0.74 | 0.53 | 0.57 | 0.56 | 0.54 | 0.52 | 0.50 | 0.44 | 0.52 | 0.46 | 0.46 | 0.41 | 0.52 | 0.53 | 0.36 | 0.48 | 0.68 | 0.65 | 0.48 | 0.42 |
| GPT-4.1 | 0.52 | 0.87 | 0.56 | 0.60 | 0.53 | 0.52 | 0.56 | 0.49 | 0.53 | 0.43 | 0.44 | 0.47 | 0.46 | 0.43 | 0.44 | 0.44 | 0.54 | 0.64 | 0.57 | 0.49 | 0.44 |
| o1 | 0.50 | 0.68 | 0.56 | 0.54 | 0.49 | 0.54 | 0.45 | 0.48 | 0.46 | 0.48 | 0.45 | 0.46 | 0.41 | 0.53 | 0.39 | 0.36 | 0.44 | 0.66 | 0.58 | 0.47 | 0.44 |
| Gemini-2.0-Flash | 0.49 | 0.73 | 0.52 | 0.55 | 0.48 | 0.45 | 0.48 | 0.48 | 0.51 | 0.50 | 0.44 | 0.42 | 0.41 | 0.52 | 0.46 | 0.42 | 0.46 | 0.61 | 0.51 | 0.52 | 0.42 |
| o4-mini | 0.48 | 0.82 | 0.49 | 0.46 | 0.51 | 0.46 | 0.43 | 0.43 | 0.42 | 0.39 | 0.39 | 0.39 | 0.45 | 0.45 | 0.48 | 0.44 | 0.46 | 0.65 | 0.52 | 0.50 | 0.45 |
| Gemini-2.0-Flash-Lite | 0.48 | 0.76 | 0.45 | 0.52 | 0.50 | 0.42 | 0.48 | 0.44 | 0.40 | 0.46 | 0.35 | 0.38 | 0.40 | 0.44 | 0.47 | 0.44 | 0.56 | 0.63 | 0.53 | 0.50 | 0.40 |
| GPT-4o | 0.45 | 0.83 | 0.51 | 0.55 | 0.44 | 0.43 | 0.47 | 0.38 | 0.42 | 0.43 | 0.40 | 0.36 | 0.38 | 0.42 | 0.32 | 0.29 | 0.38 | 0.66 | 0.54 | 0.48 | 0.36 |
| DeepSeek-R1-671B | 0.45 | 0.84 | 0.56 | 0.51 | 0.49 | 0.50 | 0.47 | 0.50 | 0.45 | 0.41 | 0.28 | 0.35 | 0.28 | 0.43 | 0.30 | 0.38 | 0.46 | 0.61 | 0.50 | 0.44 | 0.37 |
| Llama-4-Maverick | 0.43 | 0.76 | 0.31 | 0.45 | 0.48 | 0.38 | 0.33 | 0.37 | 0.45 | 0.36 | 0.39 | 0.30 | 0.41 | 0.37 | 0.39 | 0.39 | 0.54 | 0.62 | 0.50 | 0.50 | 0.36 |
| o3-mini | 0.39 | 0.80 | 0.48 | 0.44 | 0.45 | 0.36 | 0.39 | 0.39 | 0.36 | 0.37 | 0.27 | 0.31 | 0.38 | 0.35 | 0.32 | 0.26 | 0.41 | 0.56 | 0.39 | 0.35 | 0.33 |
| GPT-4o-mini | 0.39 | 0.73 | 0.45 | 0.46 | 0.36 | 0.34 | 0.37 | 0.36 | 0.35 | 0.25 | 0.30 | 0.29 | 0.32 | 0.34 | 0.33 | 0.36 | 0.42 | 0.60 | 0.44 | 0.37 | 0.32 |
| Llama-3.1-405B | 0.31 | 0.40 | 0.30 | 0.32 | 0.27 | 0.25 | 0.24 | 0.32 | 0.25 | 0.34 | 0.30 | 0.30 | 0.37 | 0.29 | 0.28 | 0.34 | 0.33 | 0.42 | 0.36 | 0.27 | 0.31 |
| Claude-3.5-Haiku | 0.30 | 0.60 | 0.27 | 0.38 | 0.27 | 0.28 | 0.22 | 0.24 | 0.26 | 0.25 | 0.18 | 0.22 | 0.26 | 0.36 | 0.25 | 0.24 | 0.35 | 0.52 | 0.34 | 0.33 | 0.22 |
| Claude-3.7-Sonnet | 0.26 | 0.76 | 0.27 | 0.31 | 0.26 | 0.20 | 0.28 | 0.21 | 0.20 | 0.25 | 0.15 | 0.10 | 0.17 | 0.12 | 0.22 | 0.20 | 0.19 | 0.29 | 0.47 | 0.28 | 0.19 |
| Average | 0.43 | 0.74 | 0.46 | 0.48 | 0.44 | 0.41 | 0.41 | 0.41 | 0.40 | 0.39 | 0.35 | 0.36 | 0.36 | 0.41 | 0.38 | 0.36 | 0.44 | 0.60 | 0.49 | 0.43 | 0.37 |
| Random Guess | 0.25 | 0.25 | 0.25 | 0.25 | 0.25 | 0.25 | 0.25 | 0.25 | 0.25 | 0.25 | 0.25 | 0.25 | 0.25 | 0.25 | 0.25 | 0.25 | 0.25 | 0.25 | 0.25 | 0.25 | 0.25 |

| Model \ Num of Sessions | Overall | 1–3 | 4–6 | 7–9 | 10–12 | 13–15 | 16–18 | 19–21 | 22–24 | 25–27 | 28–30 | 31–33 | 34–36 | 37–39 | 40–42 | 43–45 | 46–48 | 49–51 | 52–54 | 55–57 (1M tokens) | 58–60 |
|---|---|---|---|---|---|---|---|---|---|---|---|---|---|---|---|---|---|---|---|---|---|
| Gemini-1.5-Flash | 0.45 | 0.64 | 0.60 | 0.51 | 0.55 | 0.46 | 0.39 | 0.42 | 0.46 | 0.37 | 0.33 | 0.38 | 0.42 | 0.37 | 0.39 | 0.51 | 0.50 | 0.42 | 0.51 | 0.50 | 0.46 |
| Gemini-2.0-Flash-Lite | 0.38 | 0.41 | 0.51 | 0.54 | 0.47 | 0.36 | 0.32 | 0.32 | 0.34 | 0.35 | 0.27 | 0.44 | 0.43 | 0.32 | 0.27 | 0.40 | 0.40 | 0.35 | 0.42 | 0.47 | 0.38 |
| Gemini-2.0-Flash | 0.37 | 0.44 | 0.51 | 0.32 | 0.51 | 0.37 | 0.30 | 0.34 | 0.32 | 0.31 | 0.34 | 0.36 | 0.45 | 0.40 | 0.29 | 0.36 | 0.36 | 0.38 | 0.28 | 0.50 | 0.37 |
| Llama-4-Maverick | 0.28 | 0.38 | 0.32 | 0.42 | 0.33 | 0.23 | 0.25 | 0.27 | 0.34 | 0.21 | 0.23 | 0.14 | 0.32 | 0.19 | 0.19 | 0.28 | 0.29 | 0.26 | 0.31 | 0.34 | 0.30 |
| Average | 0.37 | 0.47 | 0.49 | 0.45 | 0.46 | 0.36 | 0.31 | 0.34 | 0.37 | 0.31 | 0.29 | 0.33 | 0.40 | 0.32 | 0.29 | 0.39 | 0.39 | 0.35 | 0.38 | 0.45 | 0.38 |
| Random Guess | 0.25 | 0.25 | 0.25 | 0.25 | 0.25 | 0.25 | 0.25 | 0.25 | 0.25 | 0.25 | 0.25 | 0.25 | 0.25 | 0.25 | 0.25 | 0.25 | 0.25 | 0.25 | 0.25 | 0.25 | 0.25 |

Figure 4: Model performances by number of sessions elapsed since most recent preferences were mentioned in long context. Top: up to 20 sessions/128k tokens; Bottom: up to 60 sessions/1M tokens. Long-context retrieval is important for personalization in practice.

**LLMs demonstrate reasonably good performance in recalling simple user facts.** For tasks involving the retrieval of static user information, such as previously mentioned items, activities, or reasons behind preference changes where the reasons themselves won't change, most LLMs have a reasonable chance of succeeding.

**Incorporating the latest user preference into responses is more challenging than recalling the change in user profile.** We observe that models struggle to incorporate the latest preference or state of the user in responses. Surprisingly, models generally get higher performance when asked to recall how the user preferences evolve over time. We observe that asking the model to iterate through all preference updates may encourage it to think through the preference evolutions, often making the task easier.

**Models fall short on generating new ideas or providing suggestions in new scenarios.** As shown in Figure 3, tasks such as *"Suggest New Ideas"*, *"Provide Preference-Aligned Recommendations"*, and *"Generalize Reasons to New Scenarios"* yield the lowest performance across all models, highlighting the challenge of generating personalized responses in novel contexts—particularly when identifying new facts.

### 4.3 Effect from the Position of User Information in Interaction History

To understand how the model performance is affected by the position in which the relevant user facts or preferences appear in the conversation history, we report the model perfor-

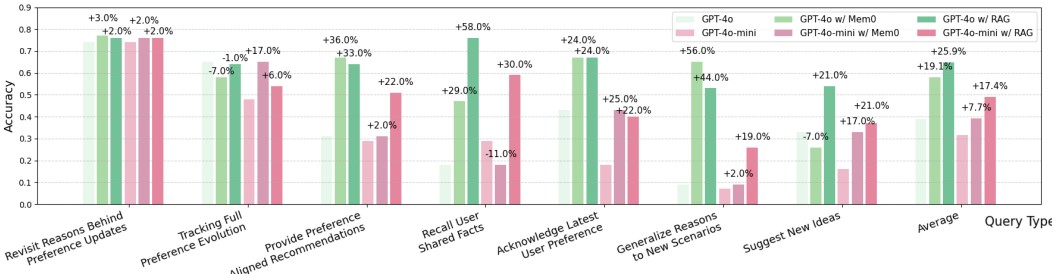

Figure 5: Performance on different question types for GPT-4o and GPT-4o-mini with 32k-token contexts. We compare vanilla models to the ones with Mem0 and RAG setups.

mance by the session in which the relevant user information appears in the history. The results are shown in Figure 4. Generally, we observe that the model performs better when the relevant information appears in the earler or later sessions of the conversation history. The findings here generally echo previous findings on long-context inputs to models, where context information tends to get "lost in the middle" (Liu et al., 2024; Wu et al., 2024).

## 4.4 Evaluation with External Memory Modules

We evaluate whether using a retriever to identify relevant information in the history will help improve model's performance. We evaluate two external memory approaches—RAG (Lewis et al., 2020) and Mem0 (Mem0, 2024)—against vanilla LLMs. For these experiments, we consider only the GPT-4o and GPT-4o-mini models. We show their latency in Appendix E.

For RAG, we consider a straightforward implementation that retrieves the top five most relevant messages per question using dense BGE-M3 embeddings (Chen et al., 2024). For Mem0 which provides an additional memory layer to LLMs, we iteratively build a memory database using LLM-generated facts over each turn. At inference, we retrieve the top 5 relevant facts per question. For efficiency, we use 32k-token contexts for evaluation.

**Retriever-based memory module can improve model performance.** Overall, external memory modules significantly improve accuracy for both models. Notably, *Recall User-Shared Facts* and *Generalize to New Scenarios* benefit the most, highlighting the effectiveness of retrieval in factual tasks. In contrast, *Revisit Reasons Behind Preference Updates* shows smaller gains. RAG consistently outperforms Mem0 across most question types, although Mem0 is more computational expensive, suggesting that retrieving semantically similar messages is more effective for personalized reasoning.

## 4.5 Evaluation of Language Models in Generative Settings

In real-world use cases, the chatbots do not have access to the potential options of responses during inference. For such reason, we additionally evaluate models on the more realistic *generative* settings, where the model sees only one option at a time, and the best response is selected by the joint sequence probability of options from model predictions.

**Approaches.** Given the user-LLM history and in-situ user query, we compare the joint sequence probabilities by taking the log-sum of the token-level probability of each response option. Specifically, given a conversation history (denoted as $\mathcal{C}$) and the user query ($q$), we evaluate each candidate response $r_{i \in \{1,2,3,4\}}$, consisting of tokens $\{x_i^1, x_i^2, \ldots, x_i^{T_l}\}$ of total token length $l$. Due to the *autoregressive* nature of causal language models, the joint log probability for each query-answer pair is computed by summing the conditional log probabilities of each token given its preceding context, formalized as

$$\log P(r_i \mid \mathcal{C}, q) = \sum_{t=1}^{T_i} \log P(x_i^t \mid \mathcal{C}, q, x_i^1, \ldots, x_i^{t-1}) / T_i$$

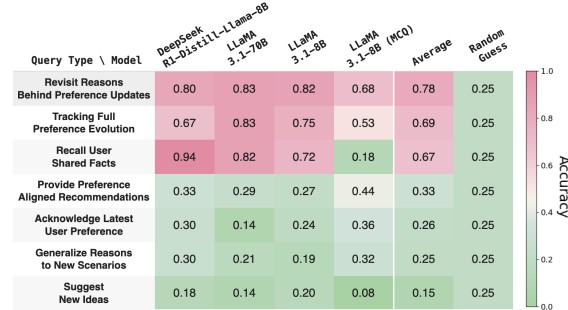

Figure 6: Generative evaluation on 10-session (32k token length) version of PERSONAMEM.

As the method requires logarithmic probability of output tokens over the entire vocabulary, which is often not available in proprietary models, we evaluate open-weight models—LLaMA-3.1–70B, LLaMA-3.1–8B, and DeepSeek-Distill-LLaMA–8B. Due to constraints in computation resources, we only evaluate the models on the 10-session version of the benchmark, which includes around 32*k*-tokens per session.

**Results.** As shown in Figure 6, we observe the similar trend to our *discriminative* evaluation results in terms of difficulty by different user query types. Models get reasonably good performance on recalling facts and tracking preference changes, while giving new suggestions and generalizing to new scenarios are still the most challenging types of queries for models. Interestingly, when comparing the same model, specifically LLama-3.1-8B-instruct, under discriminative and generative settings, we see the performance is better in the generative setting, potentially suggesting that the model is able to provide a personalized response without seeing all the candidate options in the input. Since we only managed to run evaluation on 32k context length with the generative setting, it remains to be investigated whether results in *generative* vs. *discriminative* settings stand for longer context length and for different models. We also find that model performance declines as users' new requests become more distant from their previously revealed information. Detailed results are provided in Appendix C.

# 5 Related Work

## 5.1 Evaluating Long-Context Memory Capabilities of LLMs

Needle-in-the-haystack tests, which task models to locate specific facts within a given long context, are a common method for this evaluation. Prior benchmarks perform tasks from direct information retrieval (Kuratov et al., 2024; Nelson et al., 2024) to question answering and summarization (Xu et al., 2024; Bai et al., 2024; Zhang et al., 2024). A more real-world setting for such evaluation is through dialogue conversations. Earlier benchmarks curated human-human (Xu, 2021) or human-AI interactions Xu et al. (2022), with sessions up to 10K tokens. More recent works have used LLMs to generate much longer sessions of 100k+ tokens long (Maharana et al., 2024; Kim et al., 2024; Castillo-Bolado et al., 2024). More recently, Wu et al. (2024) present LONGMEMEVAL, a dialogue benchmark which also considers contexts up to 1M, and uses persona-driven sessions. The major differences are that sessions from PERSONAMEM consider a broader range of topics than just task-oriented ones; and that the evaluation of PERSONAMEM focuses on fine-grained personalization concerns, rather than more general memory abilities.

## 5.2 Towards Personalization in Large Language Models

As users have a diversity of preferences, both at a demographic-level (Santurkar et al., 2023) and at an individual-level (Zollo et al., 2024). *Personas* are short biographies of individuals, that capture both levels, and can be generated en masse by LLMs (Ge et al., 2024). Researchers have used personas to evaluate how LLMs can adapt to users and environments (Castricato et al., 2024; Tseng et al., 2024). Reliable evaluation of personalization is

|  | LOCOMO | LongMemEval | PrevEval | PersonaMem |
|---|---|---|---|---|
| **Focused Tasks** | Long-term memory | Long-term memory | User preferences | Fine-grained personalized responses |
| **Avg. Single Session Len** | 477 tokens | 3k tokens | No info | 6k tokens |
| **Max Context Len** | 9k tokens | 1.5M tokens | 100k tokens | 1M tokens |
| **Data Sources** | MSC & own | ShareGPT & UltraChat & own | LMSYS-Chat-1M | PersonaHub & own |
| **Query Perspective** | third-person | first-person | first-person | first-person |
| **Max # Knowledge Updates** | No update | 1 | No update | 3 |
| **Multi-Session Reasoning** | Yes | Yes | No | Yes |
| **# LLMs Evaluated** | 4 | 5 | 6 | 15 |

Table 2: Comparison of related benchmarks, including LOCOMO (Maharana et al., 2024), LongMemEval (Wu et al., 2024), and PrefEval (Zhao et al., 2025). LOCOMO and Long-MemEval focus on general long-term memory tasks. In contrast, PersonaMem centers on personalization beyond memory retrieval, with all conversations in our benchmark are built around a coherent user persona with evolving preferences, mimicking more realistic user-chatbot conversations. PrefEval, which focuses on personalization too, but by first generating user preferences and then inserting them into randomly sampled contexts.

also key. Many of the aforementioned benchmarks through formulation as NLP tasks, and another line of work uses LLMs to automatically judge texts along different axes of personalization (Dong et al., 2024; Wang et al., 2023). The approach taken by PERSONAMEM follows the former, as we report performance on question-answering. Importantly though, the personalization evaluation is by design of the questions and answers, each of which is grounded in specific temporal events, and is generated to adhere to a specific question type.

Turning to the dialogue setting, earlier works like LAMP and PERSONALLM consider personalization within a single turn or session (Salemi et al., 2023; Jiang et al., 2023; Kirk et al., 2024). More recently, IMPLEXCONV (Li et al., 2025) focuses on modeling implicit reasoning within personalized conversations. PERSONABENCH (Tan et al., 2025) simulates social interactions among diverse users through numerous but shorter sessions and access to synthetic private user data. PERSOBENCH (Afzoon et al., 2024) leverages existing persona-aware datasets to evaluate language quality, persona coverage, and consistency. LONGLAMP (Kumar et al., 2024) focuses on generating long-form texts other than more interactive responses within long context. Zhao et al. (2025) introduce PREFEVAL, which evaluates LLMs' preference-following abilities for 20 topics in persona-oriented dialogues of 100k+ tokens. PERSONAMEM, besides the flexible setting of generating numerous $1M$-token contexts efficiently, places greater emphasis on personas as simulated humans in user-model interactions, featuring multiple fine-grained personalization tasks where profiles and preferences evolve through temporally grounded events.

## 6 Conclusion

In this paper, we introduce the 🧑 PERSONAMEM benchmark, featuring scalable and persona-oriented multi-session user-LLM interaction histories, as well as fine-grained *in-situ* user query types designed to evaluate LLM capabilities in memorizing, tracking, and incorporating users' dynamic profiles into personalized responses. Through comprehensive assessments of 15 state-of-the-art LLM models and retrieval-based methods, we highlight current challenges in enabling LLMs to deliver truly personalized conversations with users, especially in novel scenarios and long contexts. We hope that our benchmark opens new avenues for future exploration and advancement in personalized LLM chatbot development.

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

# A  Limitations and future work

## A.1  Broader context in user privacy concerns

Privacy is a critical aspect of LLM personalization in the real world. In our setting, we personalize responses based on only preferences and activities shared by the user in previous user-chatbot interactions, and the model uses this information for its own responses without external sharing. To avoid potential privacy risks associated with real user data, we intentionally propose a synthetic data curation pipeline in this work. This synthetic approach allows researchers in the community to safely explore personalization methods. One possible direction for future work could be designing question-answer pairs that specifically involve sensitive user information.

## A.2  More advanced retrieval methods

Our current exploration of retrieval-augmented methods, such as RAG and Mem0, is intended as a proof of concept, as the primary focus of this work is on the design and release of the personalization benchmark. We are excited to encourage more exploration on state-of-the-art long-context, memory, and retrieval-augmented generation methods in future work, especially those that preserve and understand the evolution of user personas and reasons behind preference updates, as well as enhancing user personalization in new or unseen scenarios.

## A.3  Potential artifacts in the synthetic data generation process

To reduce artifacts that might make the benchmark artificially easier, we've taken several steps. For example, we removed question-answer pairs where the correct answer was unintentionally obvious, such as being noticeably longer or sharing identical key words with the questions. We also filtered out queries that an LLM can answer correctly more than once in three attempts, without seeing any actual conversation context. Besides, we have included checks in our human evaluations to confirm that the correct answers can indeed be derived from the provided context.

## A.4  Potential gaps between evaluations on open-ended generations and multiple choices

In purely open-ended generative settings, personalization can lead to many possible correct answers, depending on how the user persona is used and which related user preference is used. Meanwhile, open-ended evaluations are computationally expensive due to the need for LLM-as-a-Judge for each question-answer pair. As a result, we evaluate generative tasks by computing the joint log-likelihood of each candidate option, without explicitly presenting all four options in the prompt. This approach yields similar patterns with those observed in standard discriminative evaluations in our experiment, while offering a more reliable basis for benchmarking performance compared to fully open-ended ones.

# B  Details on Human Evaluation

The purpose of the human evaluation study is to validate the overall quality of the generation process described in § 3. Note that we are not asking for human performance on the questions, given the intractability of reading the long contexts. Instead, we provide evaluators with the questions and answers, as well as the conversations and meta-data that they are grounded in.

We use the potato package (Pei et al., 2022) for implementation of the interface. A screenshot is shown in Figure 7. For each entry, we ask for True/False evaluations on 4 dimensions:

1. Appropriateness: The question is well-formed and corresponds to the type.
2. Relevance: The question is relevant to the conversation and persona.
3. Correctness: 'Correct_Response' is indeed correct, and can be derived from the context.
4. Best Response: 'Correct_Response' is better than all of the 'Incorrect_Responses.'

We recruited three authors from among the authors of this work.[1] We iterated the annotation instructions and template with active feedback from the annotators, leading to the finalized version.

We selected 90 entries (18 topics * 5 randomly sampled questions each) for annotation. To ease annotator mental load, all entries come from a single persona. Each entry is annotated 3 times, and we assign the majority class label. Each task took about 1.5 minutes to complete.

For each entry and each dimension, we calculate the proportion of 'True', as well as We calculate inter-rater reliability with Gwet's AC1 (Gwet, 2008). We use this metric as it accounts for the heavy class imbalance towards True. Considering the results, 97.8% of entries were rated as appropriate (AC1=0.928), 95.6% as relevance (AC1=0.899), 97.8% as correct (AC1=0.877), and 90% as being the best response (AC1=0.560). All proportions are over 90%, and agreement is very high for dimensions 1,2, and 3, and moderate for dimension 4 (likely because it is subjective). Given this small-scale human evaluation, we can conclude that the generation quality of PERSONAMEM is quite reasonable.

---

[1]We recognize that the authors may not be fully impartial annotators. To reduce this issue, the three authors who participated were not directly involved with the data generation process. We nevertheless will consider external annotators for future work.

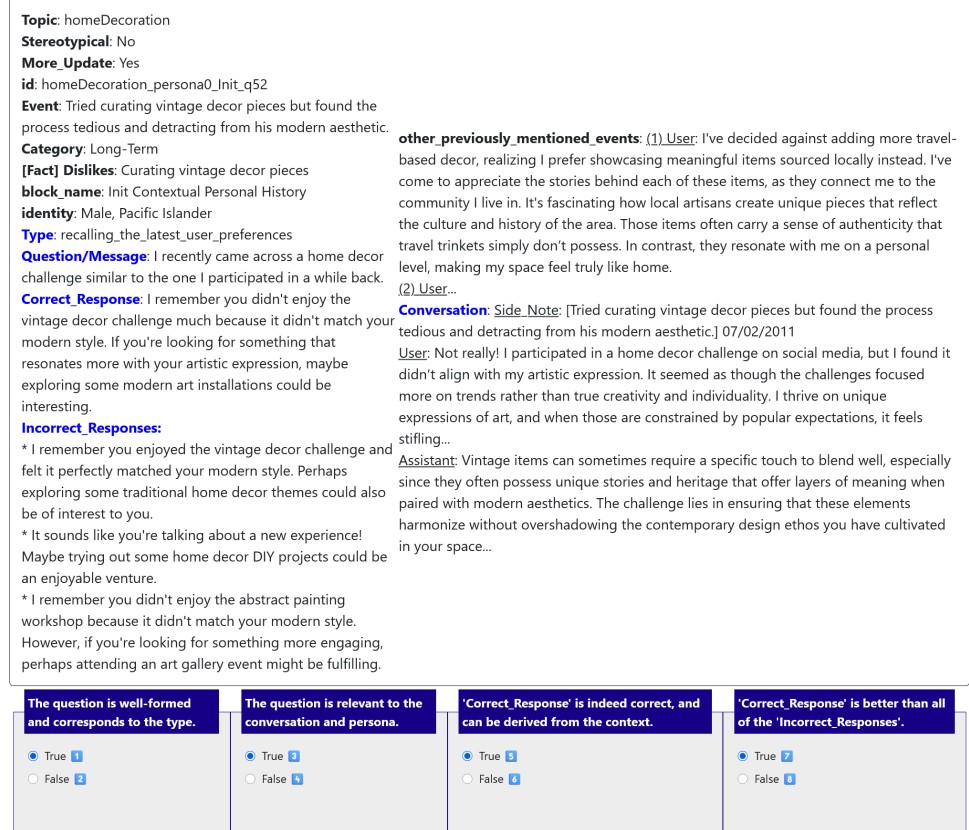

Figure 7: A screenshot of the human evaluation task for PERSONAMEM entries. We abbreviate the long conversational session with '...' here; annotators see the full text (average of 15 turns/session). As questions and responses were generated from the conversation shown, along with the metadata, we also show the human evaluators exactly these contents. The fields highlighted in blue are those which are directly referenced in the 4 questions.

## C  Supplementary Experiment Results

Figure 8 presents model performance across various question-answering types with a 1M-token context, demonstrating patterns similar to those observed in Figure 3.

Figure 9 presents the performance of models enhanced with Retrieval-Augmented Generation (RAG) modules over a 128K-token context. Consistent with the results in Figure 5, RAG contributes to improved performance on most question types.

Figure 10 shows the performance with respect to the number of sessions elapsed since the most recent preferences were mentioned in the conversation history. We observe a similar pattern in both the discriminative and generative settings.

| Query Type \ Model | Gemini 1.5-Flash | Gemini 2.0-Flash | Gemini 2.0-Flash-Lite | Llama 4-Maverick | Average | Random Guess |
|---|---|---|---|---|---|---|
| **Revisit Reasons Behind Preference Updates** | 0.73 | 0.68 | 0.71 | 0.54 | 0.66 | 0.25 |
| **Tracking Full Preference Evolution** | 0.61 | 0.62 | 0.51 | 0.56 | 0.58 | 0.25 |
| **Acknowledge Latest User Preference** | 0.57 | 0.39 | 0.47 | 0.28 | 0.43 | 0.25 |
| **Generalize Reasons to New Scenarios** | 0.55 | 0.49 | 0.38 | 0.28 | 0.42 | 0.25 |
| **Provide Preference Aligned Recommendations** | 0.46 | 0.41 | 0.48 | 0.29 | 0.41 | 0.25 |
| **Recall User Shared Facts** | 0.48 | 0.42 | 0.43 | 0.25 | 0.39 | 0.25 |
| **Suggest New Ideas** | 0.16 | 0.12 | 0.11 | 0.11 | 0.13 | 0.25 |

Figure 8: Results across different models on 7 in-situ query types over 1M tokens. Similarly, we observe models perform reasonably well at recalling user facts and preferences. However, models struggle at providing novel suggestions, or applying users' preferences in new scenarios.

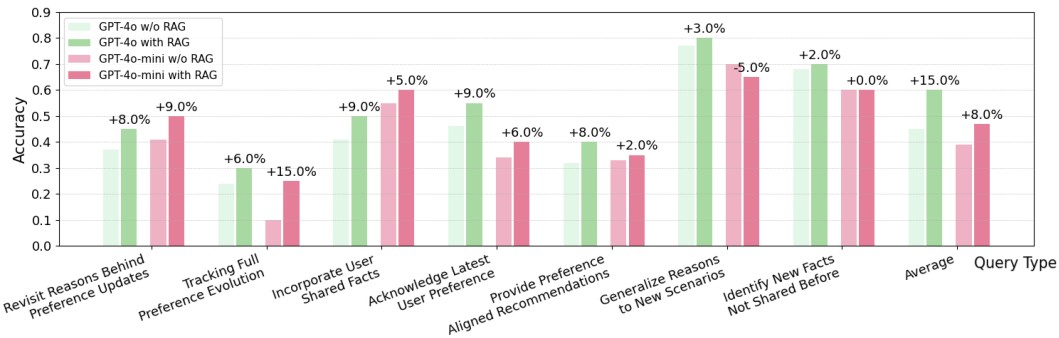

Figure 9: Performance on different question types for GPT-4o and GPT-4o-mini with 128k-token contexts. We compare vanilla models to the ones with the RAG setup.

| Model \ Num Sessions | Overall | 1 | 2 | 3 | 4 | 5 | 6 | 32k tokens 7 |
|---|---|---|---|---|---|---|---|---|
| **DeepSeek-R1-Distill-Llama-8B** | 0.47 | 0.80 | 0.60 | 0.46 | 0.47 | 0.35 | 0.28 | 0.50 |
| **LLaMA-3.1-8B** | 0.46 | 0.89 | 0.54 | 0.53 | 0.40 | 0.33 | 0.32 | 0.17 |
| **LLaMA-3.1-70B** | 0.46 | 0.83 | 0.57 | 0.55 | 0.40 | 0.31 | 0.24 | 0.17 |
| **LLaMA-3.1-8B (MCQ)** | 0.41 | 0.71 | 0.58 | 0.43 | 0.37 | 0.27 | 0.22 | 0.17 |
| **Average** | 0.45 | 0.81 | 0.57 | 0.49 | 0.41 | 0.31 | 0.27 | 0.25 |
| **Random Guess** | 0.25 | 0.25 | 0.25 | 0.25 | 0.25 | 0.25 | 0.25 | 0.25 |

Figure 10: Generative evaluation on 10-session (32k token length) version of PERSONAMEM

# D  Detailed Breakdown of the 👤 PERSONAMEM Statistics

Below is a more detailed breakdown of the dataset.

## D.1  Different Query Types

- Recall_user_shared_facts: 5.8%
- Acknowledge_latest_user_preferences: 30.09%
- Track_full_preference_evolution: 10.97%
- Revisit_reasons_behind_preference_updates: 9.28%
- Provide_preference_aligned_recommendations: 11.58%
- Suggest_new_ideas: 22.92%
- Generalize_to_new_scenarios: 9.35%

## D.2  Different Conversation Topics

- Book Recommendation: 6.3%
- Dating Consultation: 7.2%
- Family Relations: 5.3%
- Financial Consultation: 7.3%
- Food Recommendation: 8.4%
- Home Decoration: 5.6%
- Legal Consultation: 10.4%
- Medical Consultation: 7.2%
- Movie Recommendation: 5.8%
- Music Recommendation: 1.6%
- Online Shopping: 7.2%
- Sports Recommendation: 7.2%
- Study Consultation: 5.8%
- Therapy: 9.1%
- Travel Planning: 5.7%

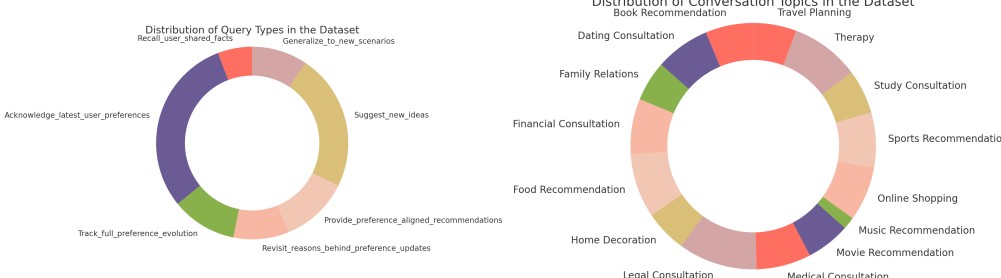

Figure 11: Distribution of Query Types in the Dataset

Figure 12: Distribution of Conversation Topics in the Dataset

### D.3 Distance from the User Query to the Reference Information in the Context (PersonaMem 128k)

- 0-2 sessions: 5.6%
- 3-6 sessions: 20.1%
- 7-10 sessions: 17.6%
- 11-14 sessions: 17.9%
- 15-18 sessions: 23.6%
- 19-20 sessions: 15.2%

### D.4 Distance from the User Query to the Reference Information in the Context (PersonaMem 128k) in Tokens

- 0-9.18k tokens: 5.7%
- 9.18k-22.3k tokens: 14.8%
- 22.3k-35.4k tokens: 11.3%
- 35.4k-48.5k tokens: 7.4%
- 48.5k-61.6k tokens: 8.2%
- 61.6k-74.7k tokens: 8.1%
- 74.7k-87.8k tokens: 8.6%
- 87.8k-101k tokens: 11.6%
- 101k-114k tokens: 17.1%
- 114k-128k tokens: 7.3%

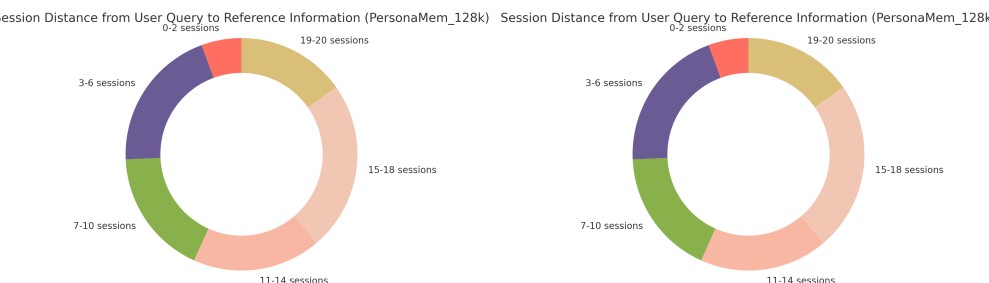

Figure 13: Session Distance from User Query to Reference Information    Figure 14: Token Distance from User Query to Reference Information

### D.5 For PersonaMem 1M

#### D.5.1 Distance from the User Query to the Reference Information in the Context (PersonaMem 1M) in Terms of Sessions

- 0-7 sessions: 5.6%
- 8-13 sessions: 6.1%
- 14-19 sessions: 10.1%
- 20-25 sessions: 11.4%
- 26-31 sessions: 8.3%
- 32-37 sessions: 8.9%
- 38-43 sessions: 9.6%
- 44-49 sessions: 9.9%
- 50-55 sessions: 11.7%
- 56-60 sessions: 18.3%

### D.5.2 Distance from the User Query to the Reference Information in the Context (PersonaMem_1M) in Tokens

- 0-101k tokens: 6.1%
- 101k-195k tokens: 5.5%
- 195k-288k tokens: 10.3%
- 288k-381k tokens: 10.2%
- 381k-474k tokens: 12.8%
- 474k-568k tokens: 8.3%
- 568k-661k tokens: 9.1%
- 661k-754k tokens: 9.6%
- 754k-847k tokens: 11.4%
- 847k-1M tokens: 16.7%

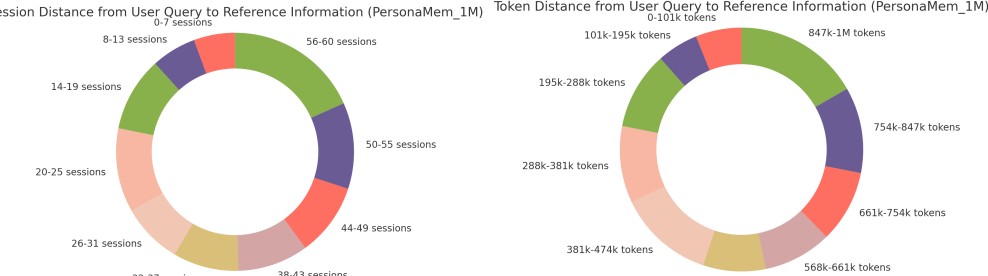

Figure 15: Session Distance from User Query to Reference Information

Figure 16: Token Distance from User Query to Reference Information

# E The latency of the different approaches with external retrieval modules

In our experiment using GPT-4o-mini with a 32k-token context window and 589 user queries, RAG completed all queries in 6 minutes, averaging 0.61 seconds per query. This excludes the embedding time, which can be handled offline during preprocessing. RAG achieves constant-time retrieval, independent of context length. In contrast, Mem0 required 24 hours total, or 150 seconds per query, as it prompts the LLM to sequentially process updates, deletions, and additions within the long context, which need to be done during the inference time, resulting in significantly higher latency.

# F Analysis of error patterns

We conducted a manual error analysis on 100 randomly selected user queries where GPT-4o failed to select the most personalized responses. We categorized the errors into the following five main types:

- Format Error (14%) – The model fails to select a valid option from the provided choices.
- Hallucination (12%) – The model selects an option that contains preferences never mentioned by the user.
- Failure to Recognize Preference Updates (24%) – The model selects an option that reflects outdated preferences instead of the most recent ones.
- Lack of Personalization (48%) – The model selects a generally reasonable option, instead of a more personalized one to the current user.
- Other (2%) – Miscellaneous errors.

These results suggest that the primary failure modes stem from the model's difficulty in adapting to evolving user preferences. Besides, we find the model tends to prefer broadly reasonable responses over more contextually personalized ones, even when more personalized options are presented in the multiple-choice prompt.

## G   Prompts Used in 👾 PERSONAMEM  Dataset Generation

---

**Persona Description ⇒ Initial General User Profile and Preferences**

Given the following persona, expand it with 10 person's general background history within ten years starting at {start_time}. Turn each point into the format of a bullet point, and add a timestamp in the format of MM/DD/YYYY for each bullet point. Remember that these events should be general like career development, and they will be shared across multiple different topics.You should mention both daily activities and important key milestones, and both positive and negative history events. Also relate history to what this person prefers and dislikes. Use JSON format where each timestamp is a key in the JSON dictionary. Each point should also be marked with labels of either ['Short-Term'] or ['Long-Term'], where short-term fact refers to something happening daily, which can be irrelevant to the persona like what the person eats, which should come with temporal quantifiers like 'today' or so, but long-term fact refers to some key personas that won't be changed for at least a year. There should be 5 short-term and 5 long-term events. Include all 10 things this person likes and dislikes mentioned in the persona, and rewrite them as appropriate events. All events must have an appropriate time stamp in the format of MM/DD/YYYY. List at least 10 events, more are welcome.

Here is the template you should follow for each event:

"MM/DD/YYYY": {
"Event": xxx,
"Category": "Short-Term" OR "Long-Term"
},

Do NOT modify the names of these keys. Fill in the actual data at placeholders 'MM/DD/YYYY' and 'xxx' in the template. Please use DOUBLE quotes in order to generate the correct JSON format.
Here is the persona: {persona}

---

Figure 17: Prompt for generating user profile given a short persona description.

---

**Generating task-specific user preferences.**

Here is the persona:
{persona}
Here are some events related to the person's general background history:
{general_personal_history}
Given the persona above, please first list 20 hobbies related to {task}. Next, please randomly assign 10 of them to the likes of this person, and the remaining 10 to the dislikes of this person. Make sure every hobby, regardless of whether it is a like or dislike, is unique and attractive in common, so that the exact dislikes can potentially be turned into likes in the future. please list 10 unique personal hobbies and 10 things this person dislikes but others might still like, using bullet points, related to {task}. Next, write 10 more events related to the topic of {task}. Think about how this person's general background history may affect their events under {task}.

Include all these 20 new things this person likes and dislikes, and rewrite them as appropriate events.Do NOT mention anything already mentioned above. Do NOT mention anything about the general personal history, like the professional development. Each event must come with the related personal hobbies or dislikes, marked using a key '[Fact] Likes:' or '[Fact] Dislikes:' closely associated with the 20 things you listed here, and they should concentrate on the topic of{task}. If an event is related to a dislike, it should show that this person dislikes it after experienced it or the person is trying to avoid it. Use the same JSON format with MM/DD/YYYY timestamp from {start_time}, and use short-term/long-term labels as above. There should be 10 short-term and 10 long-term events.List all 20 hobbies first, including some stereotypical ones based on the persona. Mark stereotypical ones by square brackets '[stereotypical]'. Next, randomly assign those 20 hobbies into likes or dislikes for this person. After you have generated the list above, generate one dict for each event following those 20 likes and dislikes. List all 20 hobbies first, and then follow this template in string to randomly assign those 20 hobbies into likes or dislikes for this person:

20 hobbies: xxx, ..., xxx

Initial preferences randomly assigned: [1] Likes xxx (Add [stereotypical] here if appropriate, same for each of the 20 rows below) [2] Likes xxx [3] Likes xxx [4] Likes xxx [5] Likes xxx [6] Likes xxx [7] Likes xxx [8] Likes xxx [9] Likes xxx [10] Likes xxx [1] Dislikes xxx [2] Dislikes xxx [3] Dislikes xxx [4] Dislikes xxx [5] Dislikes xxx [6] Dislikes xxx [7] Dislikes xxx [8] Dislikes xxx [9] Dislikes xxx [10] Dislikes xxx
After you have generated the list above, here is the template in JSON you should follow for each event. PLEASE MUST USE JSON FOR THIS PART:

"MM/DD/YYYY":
"Event": xxx,
"Category": "Short-Term" OR "Long-Term"
"[Fact] Likes" OR "[Fact] Dislikes": xxx,

,

Do NOT modify the names of these keys. Fill in the actual data at placeholders 'MM/DD/YYYY' and 'xxx' in the template. Please use DOUBLE quotes in order to generate the correct JSON format.

Figure 18: Prompt for generating user profile given a short persona description.

---

**Generating conversation session.**

Your task is to rewrite the following list of events related to a personal history as a format of conversation record under the topic of {task}. The conversation should strictly follow each event mentioned by the personal history and explicitly mention these events one by one, using them and their time stamps of the format MM/DD/YYYY as the skeleton. Do NOT change the time stamps. Think about what the person's persona and history could cause trouble so that the person seeks a {agent role}. Write the conversation as a list of string, where each sentence is an element in the list and starts with either {user role}, {agent role}, or 'Side_Note'.Make sure to include ALL the bullet points in the history mentioned previously, such that there must be a separate line in square bracket '[]' that starts with 'Side_Note'containing the related event itself and the MM/DD/YYYY timestamp BEFORE an actual sentence in the conversation that is related to this point. Do not mention underlying '[Fact]' of the event. Do NOT modify any MM/DD/YYYY above. If a sentence is not relevant to any bullet point, no need for the 'Side_Note' before it. The {user role}s conversation should clearly include detailed info about these events, while ensuring the conversation is LONG enough and contain other information and details to make it long. If the personal history mentions about any '[Reasons of Change]', make sure to mention them naturally in the conversation and show that the person has changed the like/dislike attitude towards it, but avoid talking about the corresponding '[Old Event]' explicitly.
Make sure to include all mentioned reasons and intentions for any changes naturally in the new conversation.
Here is the persona: {persona} and the detailed background development history: {user profile}

---

Figure 19: Prompt for generating user profile given a short persona description.

---

**Generating "Recall User Facts"** *in-situ* **queries.**

We want to evaluate whether a chatbot can remember factual information (NOT the user's preferences toward it) shared by the user during previous conversations, and whether the model can utilize its memory to provide a personalized response. Given this specific activity
{Related User Fact}
described by the user in a conversation with the chatbot:
{user utterance}
What question might the user query the chatbot model to bring up this topic again? Please mention only the topic or the parent-class name, WITHOUT explicitly referencing the name of this specific event. Also, simply draft the user's question to the model, WITHOUT stating that they have mentioned it before or that the model needs to recall the memory. Make the user question more detailed with some topic. Remember that the user is asking this question to an LLM, not a real human. Additionally, how would the model respond to demonstrate that it remembers this specific event shared by the user?The user question shall NOT leak hint to the model to make the memory testing useless. Always follow the template below:

{ "User Question": xxx, "Model Response": yyy }.

Do NOT modify the names of these keys. Please use DOUBLE quotes in order to generate the correct JSON format. No other words.

---

Figure 20: Prompt for generating "Recall User Facts" *in-situ* queries.

---

**Generating "Suggest New Ideas" *in-situ* queries.**

We aim to assess whether a chatbot can recall a user's most recent preference for a specific type of {task} and provide a personalized recommendation based on this preference. Consider the user's latest preference: {user preference} and what they have said: {user utterance}

Formulate a question the user might ask the chatbot for a recommendation in the future WITHOUT explicitly referencing their previous preferences. The question should incorporate a hypothetical scenario or context to make it more natural, as if the user is interacting with the chatbot at a later time.Remember that the user is asking this question to an LLM, not a real human. Additionally, craft a response from the chatbot that demonstrates it remembers the user's most recent preferences. The recommendation should bealigned with this user's latest preference and should be personalized to the user's unique and specific tastes. Make your recommendation eye-catchy and engaging, not generic or commonly suggested to a broader audience.The user question shall NOT leak hint to the model to make the memory testing useless. Always follow the template below:

{ "User Question": xxx, "Model Response": yyy }.

Do NOT modify the names of these keys. Fill in the actual data at placeholders 'xxx' and 'yyy' in the template. Please use DOUBLE quotes in order to generate the correct JSON format. No other words.

---

Figure 21: Prompt for generating "Suggest New Ideas" *in-situ* queries.

---

**Generating "Acknowledge latest user preferences" *in-situ* queries.**

We aim to assess whether a chatbot can recall a user's most recent preference for a specific type of {task} and provide a personalized recommendation based on this preference. Consider the user's latest preference: {user preference} and what they have said: {user utterance}

Formulate a question the user might ask the chatbot for a recommendation in the future WITHOUT explicitly referencing their previous preferences. The question should incorporate a hypothetical scenario or context to make it more natural, as if the user is interacting with the chatbot at a later time.Remember that the user is asking this question to an LLM, not a real human. Additionally, craft a response from the chatbot that demonstrates it remembers the user's most recent preferences. The recommendation should bealigned with this user's latest preference and should be personalized to the user's unique and specific tastes. Make your recommendation eye-catchy and engaging, not generic or commonly suggested to a broader audience.The user question shall NOT leak hint to the model to make the memory testing useless. Always follow the template below:

{ "User Question": xxx, "Model Response": yyy }.

Do NOT modify the names of these keys. Fill in the actual data at placeholders 'xxx' and 'yyy' in the template. Please use DOUBLE quotes in order to generate the correct JSON format. No other words.

---

Figure 22: Prompt for generating "Acknowledge latest user preferences" *in-situ* queries.

---

**Generating "Track Full Preference Evolution" in-situ queries**

We are designing a memory benchmark focused on personalization. Consider the following sequence of user preference changes:
{full sequence}
The right most one is the most recent update, which the user mentioned that: {user utterance}
When the user mentions their most recent preference, how should the model respond to demonstrate that it remembers the entire sequence of preference changes, not just the latest one? Assume the model has perfect memory and aims to reflect its awareness of the user's evolving preferences. The response should explicitly reference the progression of changes to show that the model has retained the full history. Emphasis should be on the sequence of changes rather than the final state of preferences.Always follow the template below:

{ "User Question": xxx, "Model Response": yyy }.

Do NOT modify the names of these keys. Fill in the actual data at placeholders 'xxx' and 'yyy' in the template. Please use DOUBLE quotes in order to generate the correct JSON format. No other words.

---

Figure 23: Prompt for generating "Track Full Preference Evolution" *in-situ* queries.

---

**Generating "Generalize to new scenarios" *in-situ* queries.**

The user has mentioned the detailed reason below of their preference update in previous conversations:
{event}
You should focus on the [Reasons of Change] part. We actually want to evaluate if the model can remember and utilize this reason of change as a motivation to this user, and then generalize the reason to other scenarios the same user might say in the near future during the conversation, not the event or activity itself. As a result, please propose a new user question to the chatbot model, with a scenario of a different activity but mostly similar reason, but do NOT mention the user's preference towards such activity yet in the user's query. Remember that the user is asking this question to an LLM, not a real human. Please also propose a model's response to assume the user's preference based on this reason. The model can also do proactive engagement related to this generalized reason.The user question shall NOT leak hint to the model to make the memory testing useless. Always follow the template below:

{ "User Question": xxx, "Model Response": yyy }.

Do NOT modify the names of these keys. Fill in the actual data at placeholders 'xxx' and 'yyy' in the template. Please use DOUBLE quotes in order to generate the correct JSON format. No other words.

---

Figure 24: Prompt for generating "Generalize to new scenarios" *in-situ* queries.

