# OpenReview forum: "Know Me, Respond to Me: Benchmarking LLMs for Dynamic User Profiling and Personalized Responses at Scale"
_colmweb.org/COLM/2025/Conference — COLM 2025_

### Official Review · Reviewer_4zES · 2025-05-07

**Rating:** 7
**Confidence:** 4
**Ethics Flag:** 1

**Summary:**

This paper introduces the PersonaMem benchmark for evaluating LLMs’ ability to use the interaction history with a user correctly to deliver personalized responses. In addition to static characteristics, a user has dynamic characteristics that change over time, and an LLM with memory should be able to track the changes and use the latest state. This paper highlights current challenges in this important feature, but it could benefit from more meaningful comparison with other works in the memory-augmented agents literature.

The benchmark dataset contains 180+ interaction histories based on 20 personas, each interaction history contains 10, 20 or 60 sessions, and each session contains 15-30 turns between the user and the LLM. The base personas are sampled from PersonaHub, the sessions span across 15 topics, and the dataset is mainly synthesized using GPT-4o. Notably, a timeline of life events (general personal history) and a series of updates under a specific topic (specific personal history) are generated to simulate evolution of the user.

12 LLMs are evaluated on 7 types of queries representing different skills. In the “long-context” setup, a full interaction history is input to the LLM. The LLMs are found to be good at recalling simple user facts, but fall short on generating new ideas or providing suggestions in new scenarios. They struggle to incorporate the latest user preference or state, but are more capable at recalling how these evolved over time. Two setups with external memory have also been evaluated: RAG and Mem0. They are both improvements, and RAG outperforms Mem0. Regardless of whether to evaluate in a discriminative setting (choosing from 4 options) or a generative setting (computing log probabilities), the results are similar.

**Reasons To Accept:**

- The dataset is a useful resource to evaluate LLMs’ capability in memorizing, tracking, and incorporating a dynamic user profile.
- The data synthesis pipeline is cost-efficient and scalable. It reuses sessions for efficiency and adds short non-personal interactions for diversity.
- The evaluation results (Figure 5) are useful for understanding current gaps and the effect of external memory on different skills, although more insights are expected e.g. on failure cases.

**Reasons To Reject:**

- The comparison with existing benchmark datasets such as LOCOMO and LongMemEval in Section 5.1 is not clear enough. It would be useful to provide concrete evidence to motivate this new dataset, e.g. a comparison table as well as examples.
- The latency of the different approaches is not measured. The efficiency benefit of external memory over inputting a full interaction history is not quantified. The paper claims that Mem0 is more computationally expensive than RAG, but no details are provided (necessary because Mem0 incurs cost to build the external memory besides retrieval cost at test time).

---

> ### Author Response · Authors · 2025-06-02
>
> Thank you for your valuable and encouraging feedback!
>
> **More detailed comparison with existing benchmarks like LOCOMO and LongMemEval**
>
> Thank you for the suggestion. We will update our related work section to include a comparison table that clarifies the differences. LOCOMO and LongMemEval focus on general long-term memory tasks, with LOCOMO framing questions from a third-person perspective rather than the user’s own. In contrast, PersonaMem centers on personalization. All conversations in our benchmark are built around a coherent user persona with evolving preferences, mimicking more realistic user-chatbot conversations. This requires models not only to remember past context but also to use the right user-specific information to generate longer, personalized responses, going beyond generally reasonable answers or simple information retrieval. We also include PrefEval, which focuses on personalization too, but by first generating user preferences and then inserting them into randomly sampled contexts.
>
> | Name        | Tasks                              | Average Session Len | Max Context Len | Data Sources for Conversations                                         | Query Perspective | Max # Knowledge Updates | Multi-Session Reasoning | # LLMs Evaluated |
> |-------------|------------------------------------|--------------|------------------|-------------------------------------------------------------------------|-------------------|-------------------------|-------------------------|------------------|
> | LOCOMO      | Long-term memory                   | 477 tokens   | 9k tokens        | MSC+their own                                                           | third-person      | No update               | Yes                     | 4                |
> | LongMemEval | Long-term memory                   | 3k tokens    | 1.5M tokens      | ShareGPT + UltraChat + their own                                       | first-person      | 1                       | Yes                     | 5                |
> | PrevEval    | User preferences                   | No info      | 100k tokens      | LMSYS-Chat-1M                                                           | first-person      | No update               | No                      | 6                |
> | PersonaMem  | Fine-grained Personalized responses | 6k tokens    | 1M tokens        | PersonaHub + We generate all persona-oriented conversations            | first-person      | 3                       | Yes                     | 15               |
>
>
>
> **The latency of the different approaches with external retrieval modules**
>
> In our experiment using GPT-4o-mini with a 32k-token context window and 589 user queries, RAG completed all queries in 6 minutes, averaging 0.61 seconds per query. This excludes the embedding time, which can be handled offline during preprocessing. RAG achieves constant-time retrieval, independent of context length.
> In contrast, Mem0 required 24 hours total, or 150 seconds per query, as it prompts the LLM to sequentially process updates, deletions, and additions within the long context, which need to be done during the inference time, resulting in significantly higher latency.

---

> > ### Comment · Reviewer_4zES · 2025-06-08
> >
> > Thank you for considering my feedback. The proposed update to the related work section would make the paper more solid in terms of the novelty of the contribution. I'm also satisfied with the additional insight into the latency of Mem0 compared to RAG. I'll increase my rating.

---

### Official Review · Reviewer_VgcQ · 2025-05-11

**Rating:** 6
**Confidence:** 4
**Ethics Flag:** 1

**Summary:**

This paper presents PERSONAMEM, a large-scale benchmark designed to evaluate the personalization capabilities of LLMs in long-context, dynamic user interactions. PERSONAMEM features over 180 user-LLM interaction histories, each with up to 60 multi-turn conversation sessions across 15 real-world tasks such as food recommendations, therapy, and travel planning. The benchmark includes 7 types of in-situ queries that test a model’s ability. The authors evaluate 12 frontier models, including GPT-4.5, Gemini 2.0, Claude, and LLaMA-3, using both discriminative (multiple-choice) and generative (log-probability ranking) settings. Results reveal significant gaps in personalization capabilities, particularly in tracking preference updates and generating responses in novel scenarios. PERSONAMEM offers a scalable and systematic framework to benchmark and analyze these challenges, with implications for future user-aware LLM design.

**Questions To Authors:**

None.

**Reasons To Accept:**

- Personalized LLMs are a key frontier in both academic and industrial NLP, and this paper addresses a major gap in standardized evaluation for this domain.
- The PERSONAMEM dataset is rich, covering dynamic personas and temporally-evolving preferences, making it more realistic and comprehensive than prior static personalization datasets.
- The use of structured prompts, chronological session generation, and multi-dimensional annotations enhances the dataset’s reliability and reproducibility.
- The paper assesses 12 major LLMs under both discriminative and generative setups, analyzing not only accuracy but also performance variations across query types and memory positioning.
- The results highlight challenges in integrating long-term user memory, particularly for novel recommendation or generalization tasks. These findings are useful for guiding future model design.

**Reasons To Reject:**

- Although the paper introduces a generative evaluation setting, it still relies on a multiple-choice framework that does not fully simulate open-ended real-world conversations. The "generative" evaluation ranks fixed candidate responses rather than assessing models’ ability to produce fully novel outputs.
- The benchmark assesses personalized response selection but does not evaluate longer chains of personalized interactions or user satisfaction over time, which are crucial for end-user deployment.
- While the results expose clear limitations of current models, the paper provides limited discussion on how models might be improved (e.g., via fine-tuning, memory augmentation, or explicit profiling techniques).

---

> ### Author Response · Authors · 2025-06-02
>
> Thank you for your valuable and encouraging feedback!
>
> **Potential gaps between evaluations on open-ended generations and multiple choices**
>
> Thanks for the suggestion. In purely open-ended generative settings, personalization can lead to many possible correct answers, depending on how the user persona is used and which related user preference is used. Meanwhile, open-ended evaluations are computationally expensive due to the need for LLM-as-a-Judge for each question-answer pair.  As a result, we evaluate generative tasks by computing the joint log-likelihood of each candidate option, without explicitly presenting all four options in the prompt. This approach yields similar patterns with those observed in standard discriminative evaluations, while offering a more reliable basis for benchmarking performance compared to fully open-ended ones.
>
> **Longer chains of personalized interactions or user satisfaction over time**
>
> We agree that user satisfactions are important considerations for real-world deployment, and these aspects are closely related to preference alignment through human feedback. However, improving user satisfaction typically requires additional data, such as user satisfaction scores that are usually sourced from humans in the real-world. Our focus in this work is on whether LLMs can utilize dynamic user personas existed in context into their responses across seven personalization tasks, in parallel to more subjective satisfaction scores. We believe that merging these two directions would lead to better end-user products in the future.
>
> **Potential solutions to improve personalization in LLMs**
>
> We appreciate the suggestions and will discuss them in our paper. While our primary goal is to benchmark existing models, we do explore some potential solutions in Section 4.4, where we present proof-of-concept experiments using external memory and retrieval modules, which improves the personalization performance and could be future directions, especially more state-of-the-art retrieval techniques or long-context tricks. It is also worthwhile to explore approaches that could enhance personalization in new or unseen scenarios and understand dynamic personas. In addition, while we assume further model fine-tuning may also improve model performance, we would consider this outcome expected, if so, and thus less informative as a research question.

---

> > ### Comment · Reviewer_VgcQ · 2025-06-11
> >
> > Thanks for the responses. I will maintain my score.

---

### Official Review · Reviewer_zZwi · 2025-05-18

**Rating:** 7
**Confidence:** 4
**Ethics Flag:** 1

**Summary:**

This paper introduces PERSONAMEM, a novel benchmark for evaluating how well large language models (LLMs) can track and respond to users' evolving traits and preferences over time. The authors present a comprehensive evaluation framework with simulated user-LLM interaction histories containing up to 60 conversation sessions across 15 real-world personalization tasks.

The quality of the technical work is good. The authors have clearly put significant effort into developing a robust benchmark that tests multiple dimensions of personalization capabilities. The data generation pipeline is well-designed, with careful consideration for creating realistic, temporally-consistent user profiles and conversations. The evaluation methodology is thorough, assessing different aspects of LLM performance across various context lengths and model architectures.

The paper is generally clear in its presentation. The methodology, benchmark design, and evaluation results are presented in a logical, easy-to-follow structure. Figure 1 effectively illustrates the benchmark's core concept, showing how user preferences evolve over time across multiple conversation sessions. The authors provide concrete examples for each of their seven query types, which helps in understanding the specific personalization capabilities being evaluated.

In terms of originality, PERSONAMEM makes a good contribution by focusing on dynamic user profiling and preference evolution over time - an area that has received limited attention in prior work. While other benchmarks have evaluated long-context understanding or static personalization, this work specifically addresses the temporal dimension of user preferences, which is crucial for real-world LLM assistants.

As LLMs become increasingly used as personal assistants, their ability to remember and adapt to evolving user preferences becomes critical for user satisfaction. The benchmark reveals significant shortcomings in current state-of-the-art models, with even frontier models like GPT-4.5 and Gemini-2.0 achieving only around 50% accuracy overall. This highlights an important area for improvement in LLM development.

**Questions To Authors:**

1. The benchmark evaluates models on their ability to select the correct response from multiple options, but real-world applications require generating appropriate responses. While this evaluation approach is reasonable, a discussion of the potential gap between discriminative and generative performance would be valuable.

2. It would be interesting to see how the performance varies across different topics/domains. Do models perform better on certain types of preferences (e.g., food) versus others (e.g., therapy)?

3. The paper briefly mentions that the benchmark contains "around 6k in-situ user query and LLM response pairs," but it would be helpful to provide a more detailed breakdown of the dataset statistics (e.g., distribution across query types, topics, etc.).

4. The authors could consider making their data generation pipeline and prompts publicly available to facilitate research on improving personalization capabilities in LLMs.

5. The paper primarily focuses on evaluating existing models rather than proposing solutions to the identified limitations. While this is perfectly acceptable for a benchmark paper, a brief discussion of potential approaches to address these limitations would strengthen the paper's impact.

**Reasons To Accept:**

1. **Novel Benchmark for an Important Capability**: PERSONAMEM addresses a critical gap in existing LLM evaluation frameworks by focusing specifically on dynamic user profiling over time. The benchmark's design is thoughtful and comprehensive, covering various aspects of personalization that matter in real-world applications.

2. **Scalable Data Generation Pipeline**: The authors introduce a well-designed, modular pipeline for generating synthetic persona-based conversations that simulate realistic preference evolution. This methodology could be valuable for other researchers exploring personalization in LLMs.

3. **Comprehensive Evaluation**: The paper evaluates 12 state-of-the-art models across multiple dimensions (7 query types, different context lengths, retrieval augmentation) providing a thorough assessment of current capabilities and limitations in this space.

4. **Practical Insights**: The findings offer valuable insights for LLM developers and researchers working on personalization. The analysis of how factors like position of information in context, retrieval techniques, and different query types affect performance provides clear directions for improvement.

5. **Rigorous Validation**: The human validation study demonstrating high agreement rates (>90% across most dimensions) provides confidence in the quality and reliability of the benchmark.

**Reasons To Reject:**

1. **Limited Exploration of Retrieval Enhancement**: While the paper includes some evaluation of retrieval-augmented approaches like RAG and Mem0, this analysis is relatively brief. A more extensive exploration of how various retrieval strategies might address the identified limitations would strengthen the paper, especially given the poor performance on longer contexts.

2. **Generative Evaluation Limitations**: The generative evaluation is conducted only on a smaller subset of the benchmark (10 sessions/32k tokens) and with fewer models due to computational constraints. A more comprehensive analysis of generation quality would be valuable, as this better reflects real-world LLM use cases.

3. **Potential Dataset Artifacts**: Although the authors conducted human validation, there could still be artifacts in the synthetic data that make the benchmark easier or harder in ways that don't reflect real user-LLM interactions. A discussion of potential limitations in the dataset generation approach would strengthen the paper.

4. **Analysis of Error Patterns**: The paper identifies which types of queries are most challenging for models, but there's limited analysis of specific error patterns or failure modes. A deeper exploration of why models fail at certain types of personalization would provide more actionable insights for improvement.

5. **Limited Discussion of Broader Context**: The paper would benefit from a more extensive discussion of how this work connects to broader questions about LLM personalization, such as the tension between privacy considerations and personalization capabilities.

---

> ### Author Response · Authors · 2025-06-02
>
> Thank you for your valuable and encouraging feedback!
>
> **Potential gaps between evaluations on open-ended generations and multiple choices**
>
> Thanks again for the suggestion. In purely open-ended generative settings, personalization can lead to many possible correct answers, depending on how the user persona is used and which related user preference is used. Meanwhile, open-ended evaluations are computationally expensive due to the need for LLM-as-a-Judge for each question-answer pair. As a result, we evaluate generative tasks by computing the joint log-likelihood of each candidate option, without explicitly presenting all four options in the prompt. This approach yields similar patterns with those observed in standard discriminative evaluations in our experiment, while offering a more reliable basis for benchmarking performance compared to fully open-ended ones.
>
> **Making the data generation pipeline and prompts publicly available to facilitate research on personalization**
>
> Yes, we will release the code, prompts, and the benchmark data to the public.
>
> **Potential solutions to improve personalization in LLMs**
>
> We will add more discussions in our paper. While our primary goal is to benchmark existing models, we do explore some potential solutions in Section 4.4, where we present proof-of-concept experiments using external memory and retrieval modules, which improves the personalization performance and could be future directions. In addition, while we assume further model fine-tuning may also improve model performance, we would consider this outcome expected, if so, and thus less informative as a research question.
>
> **More advanced retrieval methods**
>
> Our current exploration of retrieval-augmented methods, such as RAG and Mem0, is intended as a proof of concept, as the primary focus of this work is on the design and release of the personalization benchmark. We are excited to encourage more exploration on state-of-the-art long-context, memory, and retrieval-augmented generation methods in future work, especially those that preserve and understand the evolution of user personas and reasons behind preference updates, as well as enhancing user personalization in new or unseen scenarios.
>
> **Generative evaluations on longer context windows**
>
> We conduct generative evaluations on the 32k-token context due to computational limits. Computing token-level probabilities for each response requires running open-source models locally on our devices, and longer contexts often exceed their memory capacity. However, we find existing results with similar patterns to those observed in other settings with longer context windows.
>
> **Potential artifacts in the synthetic data generation process**
>
> Sure, we'll add additional discussions in our limitations section. To reduce artifacts that might make the benchmark artificially easier, we've taken several steps. For example, we removed question-answer pairs where the correct answer was unintentionally obvious, such as being noticeably longer or sharing identical key words with the questions. We also filtered out queries that an LLM can answer correctly more than once in three attempts, without seeing any actual conversation context. Besides, we have included checks in our human evaluations to confirm that the correct answers can indeed be derived from the provided context.
>
> **Deeper analysis of error patterns**
>
> We conducted a manual error analysis on 100 randomly selected user queries where GPT-4o failed to select the most  personalized responses. We categorized the errors into the following five main types:
>
> - Format Error (14%) – The model fails to select a valid option from the provided choices.
> - Hallucination (12%) – The model selects an option that contains preferences never mentioned by the user.
> - Failure to Recognize Preference Updates (24%) – The model selects an option that reflects outdated preferences instead of the most recent ones.
> - Lack of Personalization (48%) – The model selects a generally reasonable option, instead of a more personalized one to the current user.
> - Other (2%) – Miscellaneous errors.
>
> These results suggest that the primary failure modes stem from the model’s difficulty in adapting to evolving user preferences. Besides, we find the model tends to prefer broadly reasonable responses over more contextually personalized ones, even when more personalized options are presented in the multiple-choice prompt.

---

> > ### Comment · Reviewer_zZwi · 2025-06-09
> >
> > Thanks for the responses and your detailed analysis! If you could add this additional discussion in the limitations sections that would be great.

---

> ### Author Response · Authors · 2025-06-02
>
> **Performance across different conversation topics**
>
> We didn’t observe significant differences of accuracy among different topics. Question types and context window lengths matter much more significantly than conversation topics. Below is the list of average accuracy of each topic across all models and other variables.
> - Book Recommendation: 0.3729
> - Dating Consultation: 0.3402
> - Family Relations: 0.3573
> - Financial Consultation: 0.3163
> - Food Recommendation: 0.3953
> - Home Decoration: 0.3995
> - Legal Consultation: 0.3555
> - Medical Consultation: 0.3469
> - Movie Recommendation: 0.4033
> - Music Recommendation: 0.3608
> - Online Shopping: 0.3812
> - Sports Recommendation: 0.3755
> - Study Consultation: 0.3758
> - Therapy: 0.3780
> - Travel Planning: 0.3833
>
> **More detailed breakdown of the dataset statistics**
>
> Below is a more detailed breakdown and we will include charts in our manuscript.
>
> Different query types
>
> - Recall_user_shared_facts: 5.8%
> - Acknowledge_latest_user_preferences: 30.09%
> - Track_full_preference_evolution: 10.97%
> - Revisit_reasons_behind_preference_updates: 9.28%
> - Provide_preference_aligned_recommendations: 11.58%
> - Suggest_new_ideas: 22.92%
> - Generalize_to_new_scenarios: 9.35%
>
> Different conversation topics
>
> - Book Recommendation: 6.3%
> - Dating Consultation: 7.2%
> - Family Relations: 5.3%
> - Financial Consultation: 7.3%
> - Food Recommendation: 8.4%
> - Home Decoration: 5.6%
> - Legal Consultation: 10.4%
> - Medical Consultation: 7.2%
> - Movie Recommendation: 5.8%
> - Music Recommendation: 1.6%
> - Online Shopping: 7.2%
> - Sports Recommendation: 7.2%
> - Study Consultation: 5.8%
> - Therapy: 9.1%
> - Travel Planning: 5.7%
>
> Distance from the user query to the reference information in the context, in terms of sessions (PersonaMem_128k)
>
> - 0-2 sessions: 5.6%
> - 3-6 sessions: 20.1%
> - 7-10 sessions: 17.6%
> - 11-14 sessions: 17.9%
> - 15-18 sessions: 23.6%
> - 19-20 sessions: 15.2%
>
> Distance from the user query to the reference information in the context, in terms of sessions (PersonaMem_128k)
>
> - 0-9.18k tokens: 5.7%
> - 9.18k-22.3k tokens: 14.8%
> - 22.3k-35.4k tokens: 11.3%
> - 35.4k-48.5k tokens: 7.4%
> - 48.5k-61.6k tokens: 8.2%
> - 61.6k-74.7k tokens: 8.1%
> - 74.7k-87.8k tokens: 8.6%
> - 87.8k-101k tokens: 11.6%
> - 101k-114k tokens: 17.1%
> - 114k-128k tokens: 7.3%
>
> For PersonaMem_1M:
>
> Distance from the user query to the reference information in the context, in terms of sessions (PersonaMem_1M)
>
> - 0-7 sessions: 5.6%
> - 8-13 sessions: 6.1%
> - 14-19 sessions: 10.1%
> - 20-25 sessions: 11.4%
> - 26-31 sessions: 8.3%
> - 32-37 sessions: 8.9%
> - 38-43 sessions: 9.6%
> - 44-49 sessions: 9.9%
> - 50-55 sessions: 11.7%
> - 56-60 sessions: 18.3%
>
> Distance from the user query to the reference information in the context, in terms of tokens (PersonaMem_1M)
>
> - 0-101k tokens: 6.1%
> - 101k-195k tokens: 5.5%
> - 195k-288k tokens: 10.3%
> - 288k-381k tokens: 10.2%
> - 381k-474k tokens: 12.8%
> - 474k-568k tokens: 8.3%
> - 568k-661k tokens: 9.1%
> - 661k-754k tokens: 9.6%
> - 754k-847k tokens: 11.4%
> - 847k-1M tokens: 16.7%
>
>
> **Discussions in broader context such as privacy concerns**
>
> Privacy is indeed a critical aspect of LLM personalization in the real world. In our setting, we personalize responses based on only preferences and activities shared by the user in previous user-chatbot interactions, and the model uses this information for its own responses without external sharing. To avoid potential privacy risks associated with real user data, we intentionally propose a synthetic data curation pipeline in this work. This synthetic approach allows researchers in the community to safely explore personalization methods. One possible direction for future work could be designing question-answer pairs that specifically involve sensitive user information.

---

### Decision · Program_Chairs · 2025-07-08

**Decision:**

Accept

**Comment:**

This paper evaluates how effectively LLMs provide personalized assistance by constructing a benchmark to measure their capability to track evolving user profiles and respond accordingly. The benchmark is synthesized through a multi-step pipeline employing GPT-4o. The evaluation employs a multiple-choice question-answering format, requiring models to select the correct answer among four provided choices. Results demonstrate that current LLMs struggle to reliably address this task, particularly in capturing changes in user preferences over time.

Pros:
1. Given the wide adoption of LLMs as personal assistants, the central research question of whether LLMs can accurately model evolving user profiles is highly relevant and practical.
2. The paper demonstrates existing limitations of current LLMs in capturing dynamic user preferences, highlighting an important area for future research and improvement.

Cons:
1. A common concern among reviewers relates to the evaluation format. The paper evaluates models using a multiple-choice question-answering format; however, practical scenarios typically require a "generative" approach, where models must generate free-form answers without provided choices. (Note: The term "generative" here differs from the "generative setting" in the paper, where it refers to the selection method for determining the top choice.)
2. Another critical issue is the reliance on data synthesized using GPT-4o, an LLM itself. This approach may introduce biases and may not accurately reflect user interactions in the wild. Although the authors conducted human validation showing high agreement, humans only confirmed the "correct" response as superior to alternatives 90% of the time, suggesting potential quality concerns in the remaining 10%. Additionally, using an LLM-generated dataset to evaluate other LLMs inherently carries limitations.
3. Reviewers also expressed that the paper primarily focuses on evaluation without suggesting concrete avenues for improvement.

Overall, reviewers responded positively to this paper (with scores of 7, 6, and 7). This paper addresses an important and timely issue relevant to the practical deployment of LLMs as personal assistants. Despite imperfections related to the synthesized nature of the dataset, this benchmark represents a valuable starting point for further research. Therefore, I recommend acceptance of this paper.